# Evaluating ChatGPT's Performance in Generating and Assessing Dutch Radiology Report Impressions

Luc Builtjes[1]                                    LUC.BUILTJES@RADBOUDUMC.NL
Monique Brink[2]                                MONIQUE.BRINK@RADBOUDUMC.NL
Souraya Belkhir[2]                             SOURAYA.BELKHIR@RADBOUDUMC.NL
Bram van Ginneken[1]                      BRAM.VANGINNEKEN@RADBOUDUMC.NL
Alessa Hering[1,3]                             ALESSA.HERING@RADBOUDUMC.NL

[1] *Diagnostic Image Analysis Group, Radboudumc, Nijmegen, Netherlands*

[2] *Department of Radiology and Nuclear Medicine, Radboudumc, Nijmegen, Netherlands*

[3] *Fraunhofer MEVIS, Bremen, Germany*

**Editors:** Accepted for publication at MIDL 2024

## Abstract

The integration of Large Language Models (LLMs), such as ChatGPT, in radiology could offer insight and interpretation to the increasing number of radiological findings generated by Artificial Intelligence (AI). However, the complexity of medical text presents many challenges for LLMs, particularly in uncommon languages such as Dutch. This study therefore aims to evaluate ChatGPT's ability to generate accurate 'Impression' sections of radiology reports, and its effectiveness in evaluating these sections compared against human radiologist judgments. We utilized a dataset of CT-thorax radiology reports to fine-tune ChatGPT and then conducted a reader study with two radiologists and GPT-4 out-of-the-box to evaluate the AI-generated 'Impression' sections in comparison to the originals. The results revealed that human experts rated original impressions higher than AI-generated ones across correctness, completeness, and conciseness, highlighting a gap in the AI's ability to generate clinically reliable medical text. Additionally, GPT-4's evaluations were more favorable towards AI-generated content, indicating limitations in its out-of-the-box use as an evaluator in specialized domains. The study emphasizes the need for cautious integration of LLMs into medical domains and the importance of expert validation, yet also acknowledges the inherent subjectivity in interpreting and evaluating medical reports.

**Keywords:** Large Language Models, ChatGPT, Radiology Reports, Impression Generation, Fine-tuning

## 1. Introduction

The introduction of transformer models (Vaswani et al., 2017) revolutionized the field of Natural Language Processing (NLP). Among these models, Large Language Models (LLMs) like ChatGPT have demonstrated remarkable proficiency in a wide range of natural language understanding and generation tasks. They have shown to excel, in particular, in tasks involving common and well-defined language patterns, such as text completion, translation, and summarization (Chang et al., 2023).

In the field of medicine, the integration of various forms of Artificial Intelligence (AI) can enhance patient outcomes by providing support to healthcare professionals in complex decision-making processes (Topol, 2019). However, applying LLMs in the medical domain

presents unique challenges. Unlike many well-defined language tasks, medical text is filled with specialized terminology and nuanced contextual information. Nevertheless, significant effort has been put into designing LLMs specifically for such tasks (He et al., 2023), and many have already demonstrated their capabilities in the clinical field (Singhal et al., 2023; Dave et al., 2023; Rao et al., 2023; ten Berg et al., 2024).

The field of radiology in particular has shown to be a domain where AI thrives. The availability of commercial AI-based software solutions in this sector is widespread (Project AIR Working Group et al., 2024), with certain products achieving performance levels comparable to those of human radiologists (Lång et al., 2023). This advancement, coupled with the growing workload faced by radiologists, underscores the expanding influence of AI-driven automation within the realm of medical imaging (Alexander et al., 2020; Kwee and Kwee, 2021).

Central to the radiology workflow are radiology reports, serving as a communication tool between radiologists and physicians. These reports provide both a factual description and the radiologist's interpretation of imaging. A report is typically comprised of three main sections: the clinical questions, the 'Findings' section, and the 'Impression' section. The clinical questions are supplied by clinicians, outlining the reasons for conducting an imaging study. The 'Findings' section details the radiologist's observations on all organs and structures visible in the image. Finally, the 'Impression' section summarizes the most clinically relevant information from the 'Findings' section and answers the clinical questions. An example of a radiology report can be found in Appendix A.

As AI systems increasingly contribute to the generation of findings, LLMs can offer insights and interpretations to complement these automated processes. Automatic generation of 'Impression' sections exemplifies this capability. However, radiology reports pose distinctive challenges for NLP systems due to their often unstructured nature, which is compounded by the extensive use of specialized medical terminology and jargon. Particularly, automatically generating the 'Impression' section requires not only summarization, but also the ability to provide insights, merging of individual findings and offering interpretations—an inherently complex task. Moreover, the complexity deepens when reports are written in languages less commonly encountered within the NLP community, such as Dutch.

Additionally, there is a notable shift in the NLP domain towards adopting LLM-as-a-judge evaluation metrics for assessing more open-ended responses (Yuan et al., 2024; Dubois et al., 2024). Traditional objective metrics like BLEU and ROUGE scores tend to fall short in encompassing the full range of semantic information (Kaster et al., 2021), whereas Large Language Models can offer more subjective reasoning capacity (Zheng et al., 2023). The effectiveness of these novel metrics hinges on the LLMs' capacity to interpret and evaluate text in a manner similar to expert human readers. This is particularly challenging when it comes to medical reports.

In this context, this study aims to validate the concept that LLMs can serve as a integral element in the collaborative workflow between radiologist and AI. Our objective is to assess the proficiency of ChatGPT, specifically fine-tuned on a corpus of unstructured Dutch radiology reports, to generate high quality 'Impression' sections. Prior work has explored fine-tuned LLMs for non-English medical text (Liu et al., 2023), but we are the first to specifically investigate this task within the context of Dutch language.

Additionally, we examine GPT-4's capability to evaluate these sections out-of-the-box. We organized a reader study involving two physicians in radiology, instructed them with evaluating both the original and AI-generated impressions for correctness, completeness, and conciseness. These evaluations serve a dual purpose: they measure the quality of the generated content and establish a benchmark for assessing the precision of GPT-4's scoring capabilities.

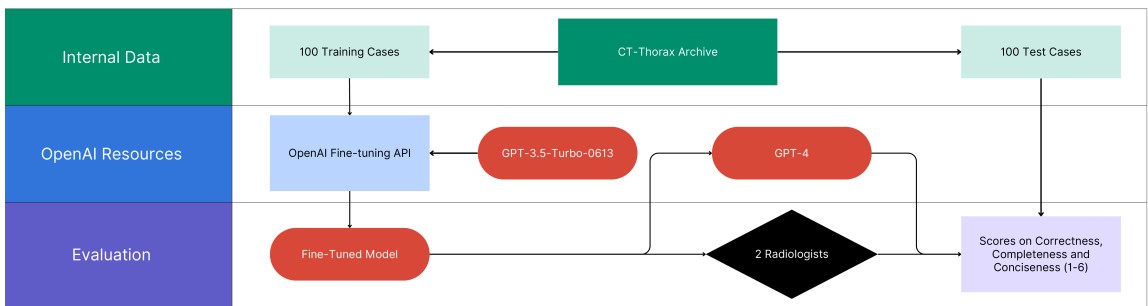

Figure 1: A schematic overview of the methodology. The flowchart shows the datasets extracted from the Radboudumc internal CT archive in green rectangles. Red ovals represent language models, and the black diamond are the two radiologists participating in the reader study.

## 2. Methods

This section will outline the methods used in our study. A schematic overview can be found in Figure 1.

### 2.1. Datasets

We sourced the data for this study retrospectively from the archive of computed tomography (CT) studies performed at the Radboudumc in Nijmegen, the Netherlands, between 2013 and 2023. We filtered the database to include only those cases that relate to thoracic scans by querying the archive metadata for CT scans of the chest with or without intravenous contrast. The clinical indications for these scans varied, reflecting a representative sample of chest CTs in an academic hospital.

To ensure data safety, every report sampled from the archive underwent iterative anonymization using our in-house anonymization software (explained in more detail in Appendix B) followed by a manual inspection. This approach was motivated by the demonstrated capability of fine-tuned models to unintentionally leak sensitive personal information from their training sets (Sun et al., 2023). Preventing such data leaks was a primary concern. Additionally, cases were excluded if they were found to contain extraneous information that could not be inferred from the 'Findings' section. For example, cases with statements such as "These findings were communicated with Physician A at timepoint B" were excluded.

Ultimately, we curated a final dataset comprising 200 reports, which was subsequently divided into both a training set and a test set, each consisting of 100 reports. This selection aligns with the recommended dataset size as outlined in OpenAI's fine-tuning guide (OpenAI, 2023) while also enabling us to guarantee complete data safety.

## 2.2. Fine-tuning of ChatGPT

Fine-tuning of LLMs involves a transfer learning process. During fine-tuning, a pre-trained model undergoes multiple training iterations on a dataset containing prompts and corresponding responses. The model's weights are updated to minimize perplexity (Jelinek et al., 1977).

In our fine-tuning process, we utilized the OpenAI fine-tuning API. It is worth emphasizing that OpenAI has not publicly disclosed the specific techniques employed within this API, presenting a challenge to scientific transparency. Numerous open-source models (Lee et al., 2020; Alsentzer et al., 2019; Touvron et al., 2023; Le Scao et al., 2022) are available for fine-tuning in a more transparent manner. However, prior testing as well as literature (Sandmann et al., 2024; Wu et al., 2023) show that these models fail to achieve the performance levels attained by ChatGPT. We determined that focusing solely on fine-tuning a state-of-the-art model represented the most optimal use of resources, both computationally and in terms of time for the participants in our reader study.

We presented each case to the API in the form of a conversation, following a predefined scenario in which a system prompt sets the context. A simulated 'user' then contributed the 'Findings' section of the report, and the 'assistant' generated and returned the 'Impression' section. The exact prompts we used are documented in Appendix C.1.

The training dataset contained a total of 55,581 tokens. We trained `GPT-3.5-turbo-0613` for 3 epochs with default settings, which took approximately 13 minutes to complete at a cost of \$1.33.

Various other transfer learning techniques such as zero-shot and few-shot learning are commonly employed in the field. However, experiments revealed that zero-shot prompting yielded suboptimal results for our specific use case, producing impressions that were highly verbose and written in mostly common Dutch, as shown in Appendix A. While inclusion of examples in the prompt did lead to a marginal improvement with respect to zero-shot learning, achieving results comparable to those of the fine-tuned model required the use of at least ten examples per case. This increased input length greatly increased inference cost while failing to produce a discernibly better output. We therefore opted not to continue further exploring this approach.

## 2.3. Reader Study

We generated the 'Impression' section for the 100 reports in the test set using our fine-tuned model by providing it the prompts presented in Appendix C.2. To evaluate the quality of these AI-generated impressions, we conducted a reader study.

For each report, the reader was presented with the 'Findings' section, followed by both the original and the generated 'Impression' section in a random order. This ensured that the reader did not know beforehand which impression section corresponded to the original or the generated version. They were instructed to evaluate each 'Impression' based on three

criteria: correctness, completeness, and conciseness. Evaluations were made using a scale from 1 to 6, with 6 indicating the highest level of quality. We provided detailed guidance for each rating category, which can be found in Appendix D.

Two human readers, M.B., a board-certified radiologist with over 14 years of clinical experience in radiology, and S.B., a resident currently specializing in radiology, participated in the reader study. Additionally, we employed GPT-4 to perform the evaluation task, providing it with the same rating guide as the radiologists. The exact prompts for this experiment are outlined in Appendix C.3.

To objectively explore the correlation between the ratings of the readers, we performed statistical analyses using the quadratically weighted Cohen's kappa coefficient $\kappa_w = 1 - \frac{\sum w_{ij} \cdot f_{o_{ij}}}{\sum w_{ij} \cdot f_{e_{ij}}}$, where $f_{o_{ij}}$ are the observed frequencies of ratings, $f_{e_{ij}}$ are the expected frequencies of ratings and $w_{ij}$ are weighting factors. This statistic allows for measurement of inter-rater reliability, taking into account the possibility of chance agreement. The coefficient can take values in the range [-1,1], where 1 means perfect agreement, 0 means agreement no better than chance, and negative values mean agreement worse than chance. Incorporating weights allows for the adjustment of penalties based on the magnitude of score differences. We specifically opted for the quadratically weighted variant of Cohen's kappa as it assigns greater weight to larger discrepancies in scores, resulting in a more nuanced evaluation of quality. We computed this metric both between human readers and GPT-4 to assess the AI's rating quality and among the human readers to evaluate inter-reader variability.

Table 1: Mean and standard deviation of the scores per reader, with 6 indicating the highest level of quality.

|  | Correctness | | Completeness | | Conciseness | |
|---|---|---|---|---|---|---|
|  | Original | Generated | Original | Generated | Original | Generated |
| M.B. | **5.72** $\pm$ 0.81 | 2.61 $\pm$ 1.52 | **5.75** $\pm$ 0.54 | 3.30 $\pm$ 1.42 | **5.31** $\pm$ 1.00 | 2.89 $\pm$ 1.66 |
| S.B. | **5.72** $\pm$ 0.75 | 3.31 $\pm$ 1.75 | **5.03** $\pm$ 1.04 | 3.20 $\pm$ 1.39 | **5.86** $\pm$ 0.53 | 4.56 $\pm$ 1.63 |
| GPT-4 | **5.64** $\pm$ 0.70 | 4.83 $\pm$ 1.41 | **4.33** $\pm$ 1.04 | 3.84 $\pm$ 1.35 | **5.81** $\pm$ 0.42 | 5.00 $\pm$ 1.22 |

## 3. Results

### 3.1. Impression Generation

Evaluation of the impression generation was conducted over the test set of 100 cases. Table 1 presents the mean scores for correctness, completeness, and conciseness as rated by each reader. The human readers, M.B. and S.B., rated the original impressions notably higher across all three metrics compared to the generated impressions.

Figure 2 graphically contrasts the performance of original versus generated impressions. These plots again show that the original impressions consistently outperformed generated

impressions. The heatmaps in figure 3 highlight the frequencies of given score combinations for each report. The preference for the original impressions is again clear.

Table 2: Weighted Cohen's kappa scores between readers using quadratic weights. Gradient colors from light to darker yellow indicate the level of agreement from weak to strong.

|  | Correctness | | Completeness | | Conciseness | |
|---|---|---|---|---|---|---|
|  | Original | Generated | Original | Generated | Original | Generated |
| M.B. vs S.B. | 0.25 | 0.53 | 0.15 | 0.56 | 0.18 | 0.39 |
| M.B. vs GPT-4 | 0.28 | 0.24 | 0.04 | 0.24 | 0.08 | 0.27 |
| S.B. vs GPT-4 | 0.28 | 0.34 | 0.21 | 0.23 | -0.03 | 0.55 |

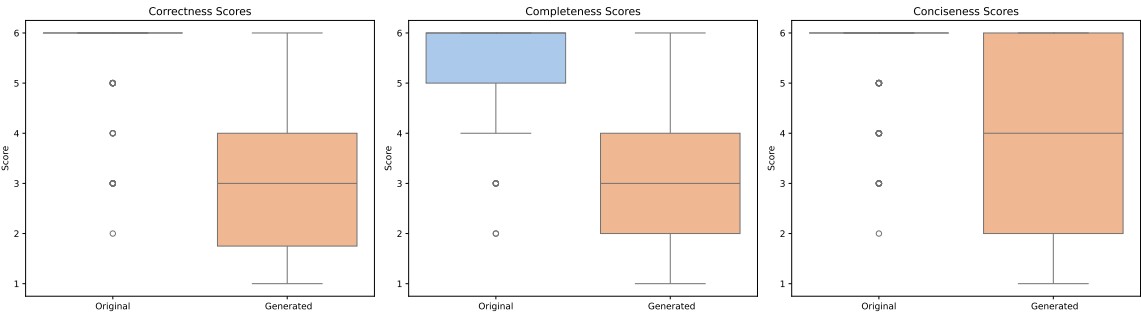

Figure 2: Box plots displaying the aggregated distribution of scores by the human readers for the three quality metrics across the original and generated impressions. For each metric, the central line represents the median score, the edges of the box indicate the interquartile range, and the whiskers show the range of the data, excluding outliers, which are plotted as individual diamonds.

### 3.2. GPT-4 Evaluation

The evaluation by GPT-4 shows less distinction between original and generated impressions. Table 1 and Figure 3 reveal that while original impressions scored comparably to human ratings, the generated impressions were rated more favorably by GPT-4. Despite this trend, the scores for the generated impressions remain lower than those for the originals.

Statistical comparisons using the weighted Cohen's kappa, summarized in Table 2, indicate a fairly weak correlation overall between readers and GPT-4. The correlation between the assessments of the human readers is shown to be relatively low as well, however.

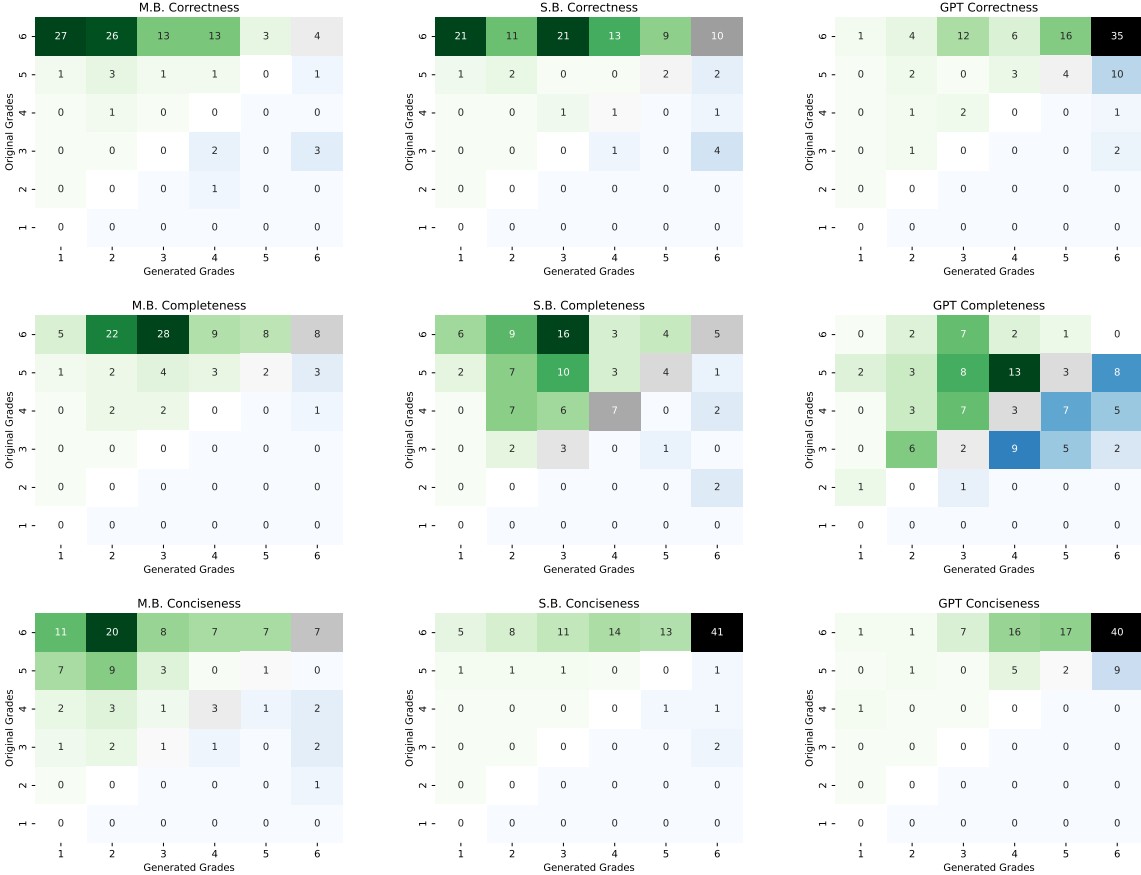

Figure 3: Heatmaps contrasting reader evaluations of original versus generated content on the three quality metrics. Shades of green indicate preference for original content, blues for generated content, and grey for equal scoring. Each cell reflects the frequency of a specific score combination.

## 4. Discussion

The findings of this study highlight the current limitations of LLMs like ChatGPT in a highly specialized domain like Dutch medical text. Despite the impressive advancements in NLP and AI, our results indicate that it remains very challenging to have LLMs generate text with the high levels of accuracy and domain-specific knowledge needed for clinical practice.

The differences seen in the ratings of correctness, completeness, and conciseness between the original and generated impressions underscore the challenge in fully automating the interpretation and summarization tasks in radiology reports. This is particularly evident in the correctness metric, where the AI-generated impressions were often marked for factual inaccuracies. Despite the model's ability to mimic the semantic style of Dutch radiologists

and generate somewhat contextually relevant content, its outputs frequently lacked the precision and reliability needed in clinical settings.

Other research has shown subpar performance of general-domain LLMs on clinical text related tasks (Hernandez et al., 2023), yet we hypothesize that the limited size of the training set was a primary reason for the suboptimal performance of our model. Fine-tuning on a larger dataset was unfeasible for our study due to the Standard Operating Procedure for data sharing we had to adhere to with regards to OpenAI, but it would likely improve performance of the fine-tuned model. However, it is important to recognize that OpenAI in their fine-tuning guide (OpenAI, 2023) suggests using a fine-tuning dataset ranging from 50 to 100 cases, a guideline which was followed in our study. This approach yielded text that appeared convincing to laypersons but revealed serious deficiencies under expert inspection. This outcome serves as a cautionary note about overreliance on such technologies without thorough validation by domain specialists.

The discrepancies in scoring patterns observed between human readers and GPT-4 highlight the limitations of deploying LLMs as evaluators within specialized fields, and shows the importance of review and validation by experts within the domain. It is important to acknowledge that our study's sample size, consisting of only two human readers, may not be sufficient to draw statistically robust conclusions. However, the consistent inclination of GPT-4 to provide higher ratings for AI-generated impressions compared to human readers remains a noteworthy observation.

One potential explanation for this discrepancy might be the experimental setup, which presented both 'Impression' sections to the human readers at the same time. This arrangement may have inadvertently prompted the readers to attempt to identify the original impression and subsequently rate it more favorably.

Moreover, the inter-reader variability observed between the human radiologists, despite all readers being provided with a detailed rating scheme, highlights the subjective nature of report interpretation and evaluation, emphasizing the need for multiple expert opinions in the clinical validation of AI-generated content.

## 5. Conclusion

This study represents a cautionary example in exploring the integration of LLMs into the medical domain. While ChatGPT demonstrated some proficiency in generating 'Impression' sections, its outputs did not consistently reach the high standards required in medical practice.

The study also revealed the limitations of using an LLM as an evaluator in specialized domains. While GPT-4's evaluations of the original impressions were somewhat aligned with human ratings, its more favorable ratings of the AI-generated impressions highlight the need for caution when employing LLMs for evaluative purposes in specialized fields. However, the observed inter-reader variability among human experts also brings to light the inherent subjectivity and complexity involved in interpreting and evaluating radiology reports. These findings emphasize the importance of critically reviewing and validating LLM-generated medical content, as well as the necessity of incorporating multiple expert opinions in this process.

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

## Appendix A.  Example of Report with Impressions Translated to English

An example of one of the reports from the test set along with the original and generated impressions in a random order. Following that is an impression generated using zero-shot prompting for comparison.

```
Relevant medical history:  Pulmonary NTM infection.  Specific question:  Follow-up
after 6 months of therapy.  Any reduction in abnormalities?

Report:  Comparison is made with the examination from <DATE>.  There is no available
scan from six months ago.

No focal pathology within the lungs.  However, there is some basodorsal
hypoventilation effects bilaterally.  Additionally, there are subpleural strands
with minimal traction on the surrounding lung parenchyma.  Very modest bronchopathic
changes perihilarly.  No evident air trapping.  Normal course of the major blood
vessels.  No noticeable calcifications in the cardiac area or vascular calcifications.
No lymphadenopathy.  A small tracheal diverticulum high on the right dorsal side.
As far as is shown in the upper abdomen, the parenchymal organs appear normal.
Regarding bony structures, apart from mild degenerative changes, there is no
evident pathology.

Impression 1:

Modest subpleural strands and hypoventilation effects are noted.  However, there
are no indications of active or specific abnormalities related to a previous
NTM infection.

Impression 2:

There are slight indications of subpleural traction bilaterally.  Perihilarly,
there is bilateral post-inflammatory bronchodilation.  There is no evidence of
current infectious pneumonia.

Zero-shot:

The imaging study reveals no significant changes compared to the previous examination
dated <DATE>.  There are no focal lung pathologies observed, although there are
basodorsal hypoventilation effects bilaterally, along with subpleural strands
exerting minimal traction on the surrounding lung tissue.  Perihilarly, there
are minor bronchopathic changes, without evident air trapping.

The major blood vessels exhibit a normal course, with no notable calcifications
in the cardiac or vascular regions, and no signs of lymphadenopathy are present.
A small tracheal diverticulum is observed high on the right dorsal side.  Parenchymal
```

```
organs in the upper abdomen appear normal.

In terms of bony structures, mild degenerative changes are noted, but no significant
pathology is evident.
```

## Appendix B. In-house Anonymizer

The in-house anonymization software used to anonymize the data for this project works in two steps. The first step is a rule-based system that has been specifically crafted to work as well as possible for the report data from the Radboudumc. It uses regular expressions to analyze n-grams (1-gram up to 5-gram) and compares these to look-up lists of names, locations and cues to replace personal health information with tags. This step is shown schematically in Figure 4. In the second step, the tags are replaced with realistic surrogates as per the "hiding in plain sight" principle (Carrell et al., 2013). This provides an additional layer of safety. Additionally, a header is prepended disclaiming that the report has been anonymized using random surrogates and that any resemblance with real persons is purely coincidental. The application of "hiding in plain sight" is outlined visually in Figure 5.

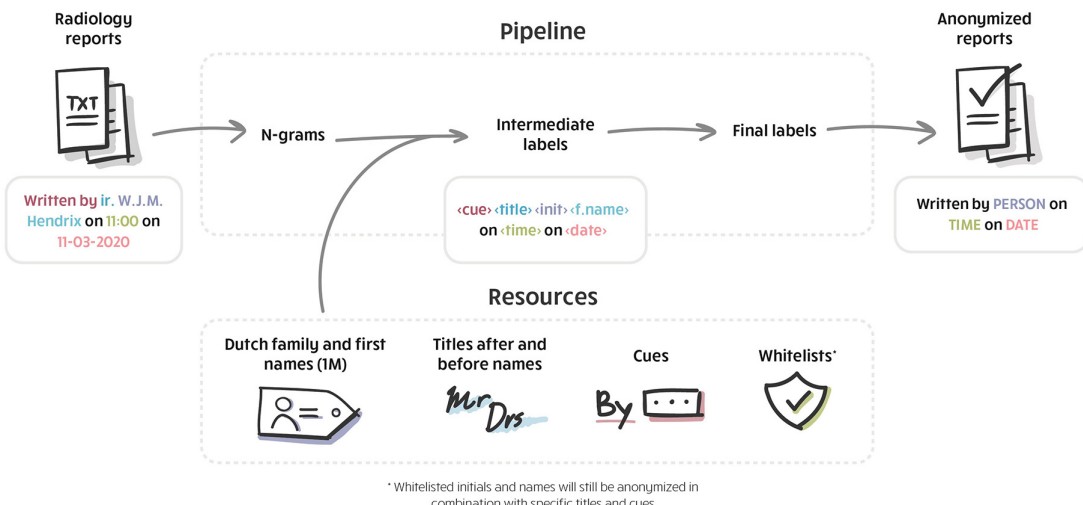

Figure 4: A schematic overview of the rule-based system used to replace personal health information with tags in the anonymization software.

## Appendix C. Prompts

This appendix outlines the prompts that we used in fine-tuning, impression generation and evaluation. It should be noted that all 'Findings' and 'Impression' sections consisted of untranslated Dutch text, while the rest of the prompts were given in English.

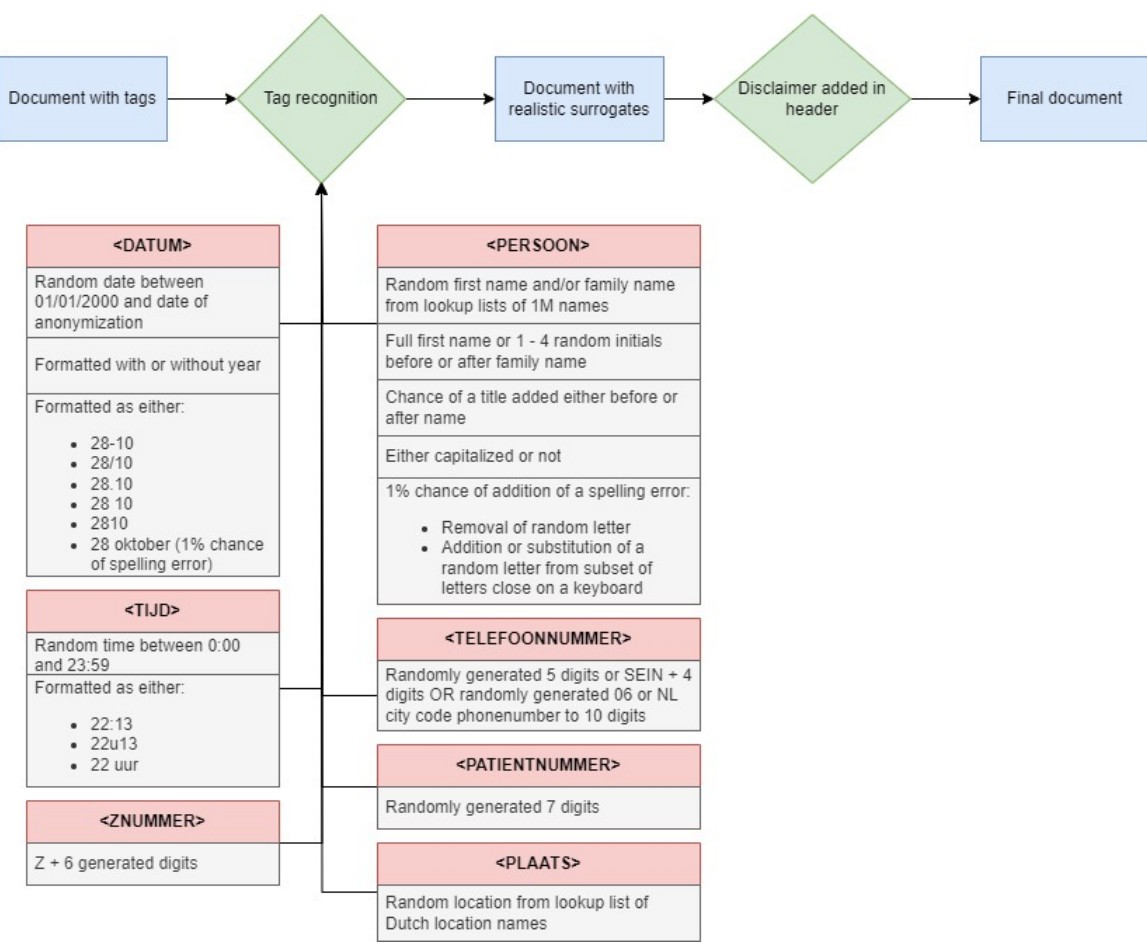

Figure 5: A schematic overview of the system applying the "hiding in plain sight" principle in the anonymization software.

### C.1. Fine-tuning prompt

```
System:  You are a Dutch radiology report generation system.  Given a set of
findings from a medical imaging report, generate the corresponding impression
section in Dutch.  Ensure that the generated text is coherent and provides a
concise summary of the findings.
User:  <Findings Section>
Assistant:  <Impression Section>
```

### C.2. Impression generation prompt

```
System:  You are a Dutch radiology report generation system.  Given a set of
findings from a medical imaging report, generate the corresponding impression
section in Dutch.  Ensure that the generated text is coherent and provides a
```

```
concise summary of the findings.
User:  <Findings Section>
```

## C.3. Evaluation prompt

```
System:  You are a Dutch radiologist.  You are helping research AI generated
radiology reports.  You are provided with the findings section of a radiology
report and two impression sections.  You must rate the impressions on correctness,
completeness and conciseness on a scale of 1-6.  Use the following rating system:
<Rating Guide>
User:  <Findings Section>
User:  Impression 1:
User:  <Impression Section 1>
User:  Impression 2:
User:  <Impression Section 2>
```

N.B. The rating guide referenced in the prompt is the one that is provided below in Appendix D.

# Appendix D. Rating Guide for the Reader Study Translated to English

## D.1. Correctness: Is the information given factually correct?

1. The impression contains serious inaccuracies that render the impression unusable and actively harmful.

2. The impression contains major incorrectnesses that render the impression unusable.

3. The impression contains some incorrectnesses in important facts making parts unusable.

4. The impression contains some incorrectnesses that lower the quality of the impression, but the important facts are correct.

5. The impression contains some incorrectnesses, but their impact is negligible.

6. The impression is completely factually correct.

## D.2. Completeness: Does the impression contain all the information you would expect given the report?

1. The impression contains only irrelevant information.

2. The impression contains some relevant points, but the most important points are absent.

3. The impression contains some of the most important points, but some are also absent.

4. The impression contains all of the main points, but some relevant side points are absent.

5. The impression is missing some possibly negligible points.

6. The impression contains all the information you would expect.

### D.3. Conciseness: Is the impression concise and contains only necessary information you would expect given the report?

1. The impression consists almost exclusively of redundant information or is even longer than the rest of the report.

2. The impression contains very much redundant information and it complicates the reading experience quite a bit.

3. The impression contains quite a lot of redundant information and it complicates the reading experience somewhat.

4. The impression contains some redundant information, but this is not enough to be very bothersome for a reader.

5. The impression contains a negligible amount of redundant information.

6. The impression contains only the necessary information.

