# OpenReview forum: "Evaluating ChatGPT's Performance in Generating and Assessing Dutch Radiology Report Impressions"
_MIDL.io/2024/Conference — MIDL 2024 Poster_

### Official Review · Reviewer_E6XC · 2024-02-22

**Confidence:** 4
**Preliminary Rating:** 2

**Summary:**

This paper explores the application of Large Language Models (LLMs) like ChatGPT in generating and evaluating 'Impression' sections of radiology reports in Dutch. It investigates whether ChatGPT can produce impressions that meet the standards of human radiologist judgments in terms of correctness, completeness, and conciseness. The study involves a dataset of CT-thorax radiology reports for fine-tuning ChatGPT and a reader study comparing the AI-generated impressions against originals using two radiologists and GPT-4. Results show that human experts rated original impressions higher, highlighting challenges in AI-generated medical text's reliability and indicating limitations in GPT-4's evaluative capabilities in specialized domains. The study emphasizes cautious LLM integration into medical domains, underscoring the need for expert validation and addressing the subjectivity in medical report interpretation and evaluation.

**Strengths:**

It is novel to apply LLM to medical texts of low-resource languages. This paper focus on very specific Dutch radiology reports and try to evaluate & generate part of the reports using ChatGPT. This approach is relatively new.

The paper is well-organized and carefully written in English.

**Weaknesses:**

There are other works introducing or reviewing how to apply LLM to medical domain. Such as 'Large Language Models Encode Clinical Knowledge' (https://arxiv.org/abs/2212.13138), 'A Survey of Large Language Models for Healthcare' (https://arxiv.org/abs/2310.05694) and 'ChatGPT in medicine: an overview of its applications, advantages, limitations, future prospects, and ethical considerations' (https://www.ncbi.nlm.nih.gov/pmc/articles/PMC10192861/). The authors failed to cite these works, which is inappropriate.

There are a lot biomedical transformer models available, such as the BioBERT (https://arxiv.org/abs/1901.08746) and ClinicalBERT (https://arxiv.org/abs/1904.03323). One can find them on Huggingface (https://huggingface.co/). Will the authors consider using some of them as the baselines in this paper? It is meaningful to compare the performance of fine-tuned ChatGPT to that of these models.

**Detailed Comments:**

This paper explores the application of Large Language Models (LLMs) like ChatGPT in generating and evaluating 'Impression' sections of radiology reports in Dutch. It investigates whether ChatGPT can produce impressions that meet the standards of human radiologist judgments in terms of correctness, completeness, and conciseness. The study involves a dataset of CT-thorax radiology reports for fine-tuning ChatGPT and a reader study comparing the AI-generated impressions against originals using two radiologists and GPT-4. Results show that human experts rated original impressions higher, highlighting challenges in AI-generated medical text's reliability and indicating limitations in GPT-4's evaluative capabilities in specialized domains. The study emphasizes cautious LLM integration into medical domains, underscoring the need for expert validation and addressing the subjectivity in medical report interpretation and evaluation.

Strength: It is novel to apply LLM to medical texts of low-resource languages. This paper focus on very specific Dutch radiology reports and try to evaluate & generate part of the reports using ChatGPT. This approach is relatively new.
The paper is well-organized and carefully written in English.

Weakness: There are other works introducing or reviewing how to apply LLM to medical domain. Such as 'Large Language Models Encode Clinical Knowledge' (https://arxiv.org/abs/2212.13138), 'A Survey of Large Language Models for Healthcare' (https://arxiv.org/abs/2310.05694) and 'ChatGPT in medicine: an overview of its applications, advantages, limitations, future prospects, and ethical considerations' (https://www.ncbi.nlm.nih.gov/pmc/articles/PMC10192861/). The authors failed to cite these works, which is inappropriate.

There are a lot biomedical transformer models available, such as the BioBERT (https://arxiv.org/abs/1901.08746) and ClinicalBERT (https://arxiv.org/abs/1904.03323). One can find them on Huggingface (https://huggingface.co/). Will the authors consider using some of them as the baselines in this paper? It is meaningful to compare the performance of fine-tuned ChatGPT to that of these models.

In general, there is no new models presented in this paper. The authors simply evaluate the performances of ChatGPT on biomedical domain datasets. Also, the evaluation is not accompanied by comparisons between other baseline models. As a result, I have to recommend weak reject to this paper.

**Justification Of The Preliminary Rating:**

In general, there is no new models presented in this paper. The authors simply evaluate the performances of ChatGPT on biomedical domain datasets. Also, the evaluation is not accompanied by comparisons between other baseline models. As a result, I have to recommend weak reject to this paper.

**Questions To Address In The Rebuttal:**

Not applicable

---

> ### Author Response · Authors · 2024-03-15
> **Rebuttal to Reviewer E6XC part 1**
>
> We would like to thank the reviewer for their insightful comments and suggestions regarding our manuscript. We appreciate the time and effort they have dedicated to evaluating our work and are eager to address the concerns raised.
>
> ***“There are other works introducing or reviewing how to apply LLM to medical domain. Such as 'Large Language Models Encode Clinical Knowledge' (https://arxiv.org/abs/2212.13138), 'A Survey of Large Language Models for Healthcare' (https://arxiv.org/abs/2310.05694) and 'ChatGPT in medicine: an overview of its applications, advantages, limitations, future prospects, and ethical considerations' (https://www.ncbi.nlm.nih.gov/pmc/articles/PMC10192861/). The authors failed to cite these works, which is inappropriate.”***
>
> We acknowledge the oversight in not citing the relevant works mentioned by the reviewer. These contributions are indeed highly relevant to our research, and we have now added these references to our paper to provide a more comprehensive background and context.
>
> ***“There are a lot biomedical transformer models available, such as the BioBERT (https://arxiv.org/abs/1901.08746) and ClinicalBERT (https://arxiv.org/abs/1904.03323). One can find them on Huggingface (https://huggingface.co/). Will the authors consider using some of them as the baselines in this paper? It is meaningful to compare the performance of fine-tuned ChatGPT to that of these models.”***
>
> We appreciate the reviewer's perspective on the potential value of using BioBERT and ClinicalBERT as baseline to compare to the performance of ChatGPT. We understand the potential value that such a comparison might bring to our study.
>
> However, we would like to provide further context on why we opted not to include these models as baselines in our current study. Our primary aim was to investigate the capabilities of state-of-the-art language generation models in performing specialized language generation tasks, specifically comparing their performance against that of human experts. While BioBERT and ClinicalBERT have demonstrated commendable performance in specific biomedical NLP tasks such as Named Entity Recognition (NER) and classification, our focus was on the domain of free text generation, where GPT-based models have consistently set the benchmark for excellence.
>
> Given this focus, we hypothesized that if ChatGPT, as one of the leading models in text generation, was unable to match the performance of human experts in this niche task, it would be unlikely for BERT-based models to surpass or equal this performance. This hypothesis is based on the inherent architectural and functional differences between BERT models, which excel in understanding and processing existing text, and GPT models, which are optimized for generating coherent and contextually relevant new text.
>
> Conducting a reader study to compare these models would indeed add value, but it would also significantly increase the duration of our study. Given the extensive time commitment already required from our readers and considering our primary research objective, we made the decision to focus on the comparison most relevant to our study's goals.
>
> We adapted the manuscript as such to include this:
>
> *In our fine-tuning process, we utilized the OpenAI fine-tuning API. It is worth empha-
> sizing that OpenAI has not publicly disclosed the specific techniques employed within this
> API, presenting a challenge to scientific transparency. Numerous open-source models (Lee
> et al., 2020; Alsentzer et al., 2019; Touvron et al., 2023; Le Scao et al., 2022) are available
> for fine-tuning in a more transparent manner. However, prior testing as well as literature
> (Sandmann et al., 2024; Wu et al., 2023) show that these models fail to achieve the per-
> formance levels attained by ChatGPT. We determined that focusing solely on fine-tuning a
> state-of-the-art model represented the most optimal use of resources, both computationally
> and in terms of time for the participants in our reader study.*

---

> ### Author Response · Authors · 2024-03-15
> **Rebuttal to Reviewer E6XC part 2**
>
> ***“In general, there is no new models presented in this paper. The authors simply evaluate the performances of ChatGPT on biomedical domain datasets. Also, the evaluation is not accompanied by comparisons between other baseline models. As a result, I have to recommend weak reject to this paper.”***
>
> We understand the reviewer's concern regarding the novelty of our work and the lack of comparative evaluation with other models. While our paper does not introduce a new model, we argue that its contribution lies in the critical assessment of ChatGPT's and GPT-4’s application to complex medical text in rare languages. This evaluation sheds light on potential weaknesses in a model that is increasingly being adopted for real-world applications, including those in sensitive fields like medicine. We emphasize the need for thorough validation by experts.
>
> We realize that we erroneously set the paper type of our submission to both “methodological development” and “validation or application”, whereas this should have been set to only the latter. This is resolved in the next iteration of our submission.

---

### Official Review · Reviewer_fM5v · 2024-02-26

**Confidence:** 4
**Preliminary Rating:** 3
**Final Rating:** 3.5

**Summary:**

This study aims to validate the concept that LLMs whether it can serve as an integral element in the collaborative workflow between radiologist and AI. More specifically, they evaluate ChatGPT's ability to generate accurate ‘Impression’ sections of radiology reports (full report includes: clinical question, summary, impressions).  For this evaluation, they fine-tune GPT-3.5 using OpenAI API, and evaluate the generated text using GPT-4 and two human radiologists.

**Strengths:**

The strength of the paper lies in the exploration of a significant research question with tangible real-world implications, utilizing a unique dataset and building upon recent advancements in the field.

**Weaknesses:**

Training set size as acknowledged by the authors.
No justification for number of epochs for training beyond that it is the default. Would this not overfit?

The paper does not provide specific insights into the justification for choosing the OpenAI API for fine-tuning. What about other large language models?

Tokenization should be considered as a key factor, at least discussed in the manuscript. It is unclear: how special characters in Dutch that are not common in English were handled; any considerations on pre- or postprocessing steps to handle Dutch-specific linguistic features effectively. This could involve adjusting how text is split into tokens or how tokenized outputs are stitched back into coherent text.

Experiments are limited to conclusively speak to the limitations of using an LLM as an evaluator in specialized domains.

The variability and subjectivity in human annotations, especially in medical contexts, are well-documented challenges. This renders the evaluations in this paper as one of its weaknesses.

**Detailed Comments:**

See the weaknesses section above.

**Justification Of Final Rating:**

The authors have thoroughly addressed my comments and made appropriate updates to the relevant sections of the paper. With some of my concerns resolved, I am pleased to increase my rating, moving towards acceptance.

**Justification Of The Preliminary Rating:**

The study conducts a relatively small number of experiments considering the significant goal of validating the use of Large Language Models (LLMs) for collaboration between radiologists and AI. Additionally, the decision to employ methods such as fine-tuning through the OpenAI API lacks a clear rationale. Nonetheless, if the authors decide to focus the study's objective to testing one particular LLM and discuss limitations of the current work, the paper could still offer valuable insights.

**Questions To Address In The Rebuttal:**

The study "Large Language Model Locally Fine-tuning (LLMLF) on Chinese Medical Imaging Reports" by Liu (2023) seems relevant to your work. Could the authors include this in the related works and discussion sections to provide a broader context?

How do the authors reconcile the acknowledged challenge of "specialized terminology and nuanced contextual information in radiology reports" with the suboptimal performance attributed to the limited training set size, especially when justifying the number of training cases based on OpenAI's generic fine-tuning guidelines?

Can the authors provide a detailed flowchart or visual representation to clarify the significant reduction from 750,000 to 200 sample reports, outlining at each stage how many data samples are reduced and the specific criteria used for these reductions?

Could the authors elaborate on the exact workings of the in-house anonymization software used in the study, to aid readers who might want to replicate this workflow?

Please provide discussions on the limitations of the OpenAI API or the potential improvements that QLoRA (since this is mentioned in the paper) might offer as an alternative approach to fine-tuning LLMs.

Please add key literature to related works section: i.e. https://arxiv.org/pdf/2302.08091.pdf

---

> ### Author Response · Authors · 2024-03-15
> **Rebuttal to Reviewer fM5v part 1**
>
> We appreciate the thoughtful comments and constructive criticism provided by the reviewer. Below, we address each concern and question raised, clarifying our methodology, choices, and the insights gained from our study.
>
> ***“The study "Large Language Model Locally Fine-tuning (LLMLF) on Chinese Medical Imaging Reports" by Liu (2023) seems relevant to your work. Could the authors include this in the related works and discussion sections to provide a broader context?”***
>
> ***“Please add key literature to related works section: i.e. https://arxiv.org/pdf/2302.08091.pdf”***
>
> We agree that the provided papers are highly relevant to our work and citing them would enhance the manuscript. We have now included these references.
>
> ***“Can the authors provide a detailed flowchart or visual representation to clarify the significant reduction from 750,000 to 200 sample reports, outlining at each stage how many data samples are reduced and the specific criteria used for these reductions?”***
>
> We acknowledge a lack of clarity in our initial explanation of the data gathering process. The dataset with 750,000 cases is a local archive containing all data from CT studies from Radboudumc (2000-2023) in a non-anonymized form. We did not necessarily perform a reduction to 200 samples, but more so sampled from this archive until we reached our dataset size of 200 cases. This sampling involved iterative anonymization steps, both manual and automatic. Additionally, we were bound to compliance with Radboudumc's Standard Operating Procedures for data sharing, which does not view OpenAI as a trusted partner. Given OpenAI's guidelines and our data privacy commitments, we found this sample size to be optimal for guaranteeing full data safety, while still being able to gain meaningful results and insights with our fine-tuning experiment.
>
> We have revised our manuscript to provide a clearer, more detailed account of this process as such:
>
> *We sourced the data for this study retrospectively from the archive of computed tomography
> (CT) studies performed at the Radboudumc in Nijmegen, the Netherlands, between 2013
> and 2023. We filtered the database to include only those cases that relate to thoracic scans
> by querying the archive metadata for CT scans of the chest with or without intravenous
> contrast. The clinical indications for these scans varied, reflecting a representative sample
> of chest CTs in an academic hospital.
> To ensure data safety, every report sampled from the archive underwent iterative anonymiza-
> tion using our in-house anonymization software (explained in more detail in Appendix B)
> followed by a manual inspection. This approach was motivated by the demonstrated ca-
> pability of fine-tuned models to unintentionally leak sensitive personal information from
> their training sets (Sun et al., 2023). Preventing such data leaks was a primary concern.
> Additionally, cases were excluded if they were found to contain extraneous information that
> could not be inferred from the ’Findings’ section. For example, cases with statements such
> as “These findings were communicated with Physician A at timepoint B” were excluded.
> Ultimately, we curated a final dataset comprising 200 reports, which was subsequently divided into both a training set and a test set, each consisting of 100 reports. This selection aligns with the recommended dataset size as outlined in OpenAI's fine-tuning guide (OpenAI, 2023) while also enabling us to guarantee complete data safety.*

---

> > ### Comment · Reviewer_fM5v · 2024-03-26
> > **Response to Rebuttal part 1**
> >
> > Thank you for your detailed response clarifying data sampling process.

---

> ### Author Response · Authors · 2024-03-15
> **Rebuttal to Reviewer fM5v part 2**
>
> ***“How do the authors reconcile the acknowledged challenge of "specialized terminology and nuanced contextual information in radiology reports" with the suboptimal performance attributed to the limited training set size, especially when justifying the number of training cases based on OpenAI's generic fine-tuning guidelines?”***
>
> As mentioned in the previous point, our data sharing agreement formed a limiting factor with regards to how much fine-tuning we were able do. We agree that a larger dataset could enhance model performance. However, it is notable that our limited dataset yielded results that were promising to the untrained eye, highlighting the importance of expert validation regarding the use of LLMs in handling niche, complex tasks. This experiment has value as a case study on cautious LLM interpretation, both for text generation as well as evaluation.
>
> We expanded our discussion section to more specifically mention the likeliness that better model performance is possible with a bigger training set:
>
> *We hypothesize that the limited size of the training set was a primary reason for the suboptimal performance of the model. Fine-tuning on a larger dataset was unfeasible for our study due to the Standard Operating Procedure for data sharing we had to adhere to with regards to OpenAI, but it would likely improve performance of the fine-tuned model. However, it is important to recognize that OpenAI in their fine-tuning guide (OpenAI, 2023) suggests using a fine-tuning dataset ranging from 50 to 100 cases, a guideline which was followed in our study. This approach yielded text that appeared convincing to laypersons but revealed serious deficiencies under expert inspection. This outcome serves as a cautionary note about overreliance on such technologies without thorough validation by domain specialists.*
>
>
>
> ***“Could the authors elaborate on the exact workings of the in-house anonymization software used in the study, to aid readers who might want to replicate this workflow?”***
>
> Our in-house anonymization software works in two steps. The first step is a rule-based system that has been specifically crafted to work as well as possible for the report data from the Radboudumc. It uses regular expressions and look-up lists of names and locations to replace the personal health information with tags. In the second step, the tags are replaced with realistic surrogates as per the “hiding in plain sight” principle [1]. This provides an additional layer of safety.
>
> We agree that it is beneficial to be as transparent as possible regarding the procedure. We therefore added a section to the Appendix outlining our anonymization method in more detail.
>
> ***“Please provide discussions on the limitations of the OpenAI API or the potential improvements that QLoRA (since this is mentioned in the paper) might offer as an alternative approach to fine-tuning LLMs.”***
>
> We recognize the oversight in not thoroughly discussing the OpenAI API's limitations. The manuscript was updated to address this aspect more comprehensively and provide alternative options. Furthermore, we have opted to remove the mention of QLoRA to prevent confusion. The section now reads as follows:
>
> *In our fine-tuning process, we utilized the OpenAI fine-tuning API. It is worth em-
> phasizing that OpenAI has not publicly disclosed the specific techniques employed within
> this API, presenting a challenge to scientific transparency. Numerous open-source models
> (Lee et al., 2020; Alsentzer et al., 2019; Touvron et al., 2023; Le Scao et al., 2022) are available for fine-tuning in a more transparent manner. However, prior testing as well as literature (Sandmann et al., 2024; Wu et al., 2023) show that these models fail to achieve the performance levels attained by ChatGPT. We determined that focusing solely on fine-tuning a
> state-of-the-art model represented the most optimal use of resources, both computationally
> and in terms of time for the participants in our reader study.*
>
>
>
> [1] Carrell, D., Malin, B., Aberdeen, J., Bayer, S., Clark, C., Wellner, B., & Hirschman, L. (2013). Hiding in plain sight: use of realistic surrogates to reduce exposure of protected health information in clinical text. Journal of the American Medical Informatics Association, 20(2), 342-348.

---

> > ### Comment · Reviewer_fM5v · 2024-03-26
> > **Response to Rebuttal part 2**
> >
> > Thank you for your detailed response addressing our concerns.

---

### Official Review · Reviewer_swSk · 2024-02-29

**Confidence:** 4
**Preliminary Rating:** 3
**Recommendation:** Poster
**Final Rating:** 3.5

**Summary:**

Authors propose a study on evaluating Large Language Models, more specifically ChatGPT, performance to generate and assess radiology report impression generation from radiology report findings in Dutch. The approach is based on fine-tuning GPT-3.5 model to generate these reports on 100 training cases and evaluating the results on 100 testing cases. The analysis uses correctness, completeness, and conciseness scores (0-6) between 2 radiology experts, GPT-4 model, and the original and generated reports. The study demonstrates that ChatGPT offers reports with consistently lower scores, as well as GPT-4 to rate more favorably generated reports.

**Strengths:**

- Paper is very well-written, organization is clear;
- Related work is adequately addressed, and the study has a clear clinical relevance and potential impact;
- Study methodology and evaluation are coherent and precise, as well as reproducible (minus the use of private data and necessary human evaluation)

**Weaknesses:**

- Regarding the fact that this is not a methodological paper, I would expect such a study to integrate more testing cases, models, and more in-depth analysis (see major comments).
- From an "archive [...] of over 750,000 thoracic and abdominal computed tomography (CT) volumes and their respective radiology reports", using only 200 reports (100 for training, 100 for testing) appears as quite dramatic and drastic cut the potential of the database.

**Detailed Comments:**

Major comments:
- There is a big concern about the drastic cut to obtain only 200 reports from the 750k initial dataset. Also considering that training data size to needed to fine-tune such models could be an interesting analysis;
- The methodology used finetuning+testing only; other transfer / evaluation techniques such as zero-shot, one-shot, few-shot, chain-of-thoughts, etc. could be used with LLMs and would be highly interesting to apprehend their performances;
- Only ChatGPT model was, and more particularly fine tuning in a black box way ("utilized the OpenAI fine-tuning API. It is, however, worth emphasizing that OpenAI has not publicly disclosed the specific techniques employed within this API"). Several open sourced LLMs exist (Llama, Bloom, etc.) and could also be tested. This comment also holds for the model used for assessing the results, which is GPT-4 only.
- Using open-sourced models could also benefit from using different fine-tuning strategies and it would also be interesting to investigate these;
- Using a dataset in a different language is indeed interesting, but the use of public English datasets could also be considered.

Minor comments:
- Figure 1: detail caption; text could be slightly bigger to me
- Tables: maybe highlight 1st, 2nd best scores with bold, underlined?

**Justification Of Final Rating:**

During the rebuttal period, authors provided clarification to the context and evaluation process of this study.
Nevertheless, as all reviewers seem to agree, this study lacks broader perspectives and comparisons to the existing models and approaches using LLMs, particularly considering that this is a “validation or application” submission.
Hence I would maintain my rating to borderline, on the accept side, acknowledging the strengths of this paper.

**Justification Of The Preliminary Rating:**

Overall, this paper is very well-written and clear. It offers an evaluation study of LLMs in a highly clinically relevant setting. However I am not sure that the study in its actual form provides enough experiments and information.

**Questions To Address In The Rebuttal:**

- Concern about the reason behind obtaining only 200 reports;
- Expanding the study with suggestions in the detailed comments would greatly benefit the study, or at least further discuss these points.

**Special Issue:**

No

---

> ### Author Response · Authors · 2024-03-15
> **Rebuttal to Reviewer swSk part 1**
>
> We appreciate the reviewers feedback on our paper. Their comments have helped highlight the paper's strengths and, more importantly, the areas that require clarification and improvement. We address each point as follows:
>
> ***“There is a big concern about the drastic cut to obtain only 200 reports from the 750k initial dataset. Also considering that training data size to needed to fine-tune such models could be an interesting analysis;”***
>
> We acknowledge the concern regarding the reduction to 200 reports from an initial dataset of 750,000. It's important to clarify that the 750,000 encompasses all data gathered from all CT studies performed at Radboudumc from 2000 to 2023, which include unanonymized reports alongside CT images. Ensuring data privacy necessitated the use of our in-house anonymization software, followed by manual verification to remove any personal health information that was missed. This process significantly limited the speed at which we could prepare the data. Furthermore, adherence to Radboudumc's Standard Operating Procedure for external data sharing, which does not recognize OpenAI as a trusted partner, mandated extra precautions for data privacy. As such, we opted for a dataset size of 100 reports, aligning with OpenAI's recommendations for fine-tuning while ensuring complete data privacy.
>
> We realize that the way the ‘Datasets’ section of our paper was structured was not optimal and led to confusion. We updated it as follows to solve this:
>
> *We sourced the data for this study retrospectively from the archive of computed tomography
> (CT) studies performed at the Radboudumc in Nijmegen, the Netherlands, between 2013
> and 2023. We filtered the database to include only those cases that relate to thoracic scans
> by querying the archive metadata for CT scans of the chest with or without intravenous
> contrast. The clinical indications for these scans varied, reflecting a representative sample
> of chest CTs in an academic hospital.
> To ensure data safety, every report sampled from the archive underwent iterative anonymiza-
> tion using our in-house anonymization software (explained in more detail in Appendix B)
> followed by a manual inspection. This approach was motivated by the demonstrated ca-
> pability of fine-tuned models to unintentionally leak sensitive personal information from
> their training sets (Sun et al., 2023). Preventing such data leaks was a primary concern.
> Additionally, cases were excluded if they were found to contain extraneous information that
> could not be inferred from the ’Findings’ section. For example, cases with statements such
> as “These findings were communicated with Physician A at timepoint B” were excluded.
> Ultimately, we curated a final dataset comprising 200 reports, which was subsequently divided into both a training set and a test set, each consisting of 100 reports. This selection aligns with the recommended dataset size as outlined in OpenAI's fine-tuning guide (OpenAI, 2023) while also enabling us to guarantee complete data safety.*
>
> ***“The methodology used finetuning+testing only; other transfer / evaluation techniques such as zero-shot, one-shot, few-shot, chain-of-thoughts, etc. could be used with LLMs and would be highly interesting to apprehend their performances;”***
>
> We agree that exploring various transfer and evaluation methodologies, such as zero-shot, one-shot, few-shot, and chain-of-thought, offers valuable insights into model performance. Initial explorations with these methodologies on ChatGPT were conducted. We found that zero-shot Impressions consisted generally of extensive lists containing most of the 'Findings' section, written in mostly common Dutch. Few-shot prompting showed promise but only started to approach the level of the fine-tuned model with 10+ examples, which also greatly increased inference cost due to the increase in token count per example. Including the few-shot examples would also increase the already lengthy reading time of our reader study. Considering these factors, we decided to prioritize fine-tuning.
>
> We edited the manuscript, so it explains this decision-making process like so:
>
> *Various other transfer learning techniques such as zero-shot and few-shot learning are commonly employed in the field. However, experiments revealed that zero-shot prompting yielded suboptimal results for our specific use case, producing impressions that were highly verbose and written in mostly common Dutch. While inclusion of examples in the prompt did lead to a marginal improvement with respect to zero-shot learning, achieving results comparable to those of the fine-tuned model required the use of at least ten examples per case. This increased input length greatly increased inference cost while failing to produce a discernibly better output. We therefore opted not to continue further exploring this approach.*

---

> > ### Author Response · Authors · 2024-03-15
> > **Rebuttal to Reviewer swSk part 2**
> >
> > ***“Only ChatGPT model was, and more particularly fine tuning in a black box way ("utilized the OpenAI fine-tuning API. It is, however, worth emphasizing that OpenAI has not publicly disclosed the specific techniques employed within this API"). Several open sourced LLMs exist (Llama, Bloom, etc.) and could also be tested. This comment also holds for the model used for assessing the results, which is GPT-4 only.”***
> >
> > We acknowledge the critique regarding our exclusive use of ChatGPT and OpenAI's fine-tuning API, particularly the lack of transparency about its underlying techniques. Our ongoing research aims to explore open-source LLMs like Llama as potential alternatives. However, our current findings indicate that these models do not yet match the performance of OpenAI's models, especially for specialized tasks like generating radiology report impressions in Dutch. This is backed up by literature [1,2]. This project aimed to benchmark the state-of-the-art against human performance in this niche area. We felt like it would not be the best use of the time of the readers in our study to have them judge outputs from worse performing models.
> >
> > We have revised the manuscript to reflect our rationale for selecting ChatGPT and the decision against incorporating additional models into our study as such:
> >
> > *In our fine-tuning process, we utilized the OpenAI fine-tuning API. It is worth em-
> > phasizing that OpenAI has not publicly disclosed the specific techniques employed within
> > this API, presenting a challenge to scientific transparency. Numerous open-source models
> > (Lee et al., 2020; Alsentzer et al., 2019; Touvron et al., 2023; Le Scao et al., 2022) are available for fine-tuning in a more transparent manner. However, prior testing as well as literature (Sandmann et al., 2024; Wu et al., 2023) show that these models fail to achieve the performance levels attained by ChatGPT. We determined that focusing solely on fine-tuning a
> > state-of-the-art model represented the most optimal use of resources, both computationally
> > and in terms of time for the participants in our reader study.*
> >
> >
> >
> > ***“Using open-sourced models could also benefit from using different fine-tuning strategies and it would also be interesting to investigate these;”***
> >
> > While we acknowledge the interest in exploring diverse fine-tuning strategies for open-source models, such investigation was beyond this project's scope. However, we agree that it presents a promising direction for future research.
> >
> > ***“Using a dataset in a different language is indeed interesting, but the use of public English datasets could also be considered.”***
> >
> > The choice to utilize a Dutch dataset was deliberate, aimed at assessing the capabilities of LLMs in processing Dutch medical texts. While the efficacy of LLMs with English medical datasets is well-documented [3] our objective was to determine if comparable performance extends to Dutch.
> >
> > References to studies utilizing English datasets have been added to the manuscript, highlighting the contrast in performance between those studies and ours.
> >
> > ***“Figure 1: detail caption; text could be slightly bigger to me”***
> >
> > We have updated the caption to be more detailed. We also increased the font size of the text in the figure.
> >
> > ***“Tables: maybe highlight 1st, 2nd best scores with bold, underlined?”***
> > We have implemented the reviewers feedback by highlighting the best scores per category between the original and generated impressions for Table 1. For Table 2, we have color coded different ranges of the kappa score to increase readability.
> >
> > [1] Sandmann, S., Riepenhausen, S., Plagwitz, L., & Varghese, J. (2024). Systematic analysis of ChatGPT, Google search and Llama 2 for clinical decision support tasks. Nature Communications, 15(1), 2050.
> >
> > [2] Wu, S., Koo, M., Blum, L., Black, A., Kao, L., Scalzo, F., & Kurtz, I. (2023). A comparative study of open-source large language models, gpt-4 and claude 2: Multiple-choice test taking in nephrology. arXiv preprint arXiv:2308.04709.
> >
> > [3] Van Veen, D., Van Uden, C., Blankemeier, L., Delbrouck, J. B., Aali, A., Bluethgen, C., ... & Chaudhari, A. S. (2024). Adapted large language models can outperform medical experts in clinical text summarization. Nature Medicine, 1-9.

---

> > > ### Comment · Reviewer_swSk · 2024-03-25
> > > **Response to rebuttal**
> > >
> > > Thank you for the detailed answers to the reviews and the proposed modifications to the manuscript.
> > > Clarifications were made to understand the context of this study and its relevance better.

---

### Meta-Review · Area_Chair_vzcm · 2024-04-04

**Recommendation:** Accept (Poster)
**Confidence:** 4

**Metareview:**

While the reviewers raised issues about limited dataset size and justification for the choice of LLM, on balance they found the paper clear and the evaluation convincing. They also stressed the importance of the topic for the medical field.

---

### Decision · Program_Chairs · 2024-04-06

Accept (Poster)